# Regularized Langevin Dynamics for Combinatorial Optimization

**Shengyu Feng** [1]  **Yiming Yang** [1]

## Abstract

This work proposes a simple yet effective sampling framework for combinatorial optimization (CO). Our method builds on discrete Langevin dynamics (LD), an efficient gradient-guided generative paradigm. However, we observe that directly applying LD often leads to limited exploration. To overcome this limitation, we propose the *Regularized Langevin Dynamics (RLD)*, which enforces an expected distance between the sampled and current solutions, effectively avoiding local minima. We develop two CO solvers on top of RLD, one based on simulated annealing (SA), and the other one based on neural network (NN). Empirical results on three classic CO problems demonstrate that both of our methods can achieve comparable or better performance against the previous state-of-the-art (SOTA) SA- and NN-based solvers. In particular, our SA algorithm reduces the runtime of the previous SOTA SA method by up to 80%, while achieving equal or superior performance. In summary, RLD offers a promising framework for enhancing both traditional heuristics and NN models to solve CO problems. Our code is available at https://github.com/Shengyu-Feng/RLD4CO.

## 1. Introduction

Combinatorial Optimization (CO) problems are central challenges in computer science and operations research (Papadimitriou & Steiglitz, 1998), with diverse real-world applications such as logistics optimization (Chopra & Meindl, 2001), workforce scheduling (Ernst et al., 2004), financial portfolio management (Rubinstein, 2002; Lobo et al., 2007), distributed computing (Zheng et al., 2022; Feng et al., 2025a), and bioinformatics (Gusfield, 1997). Despite their wide-ranging utility, CO problems are inherently difficult

due to their non-convex nature and often NP-hard complexity, making them intractable in polynomial time by exact solvers. Traditional CO algorithms often rely on hand-crafted domain-specific heuristics, which are costly and difficult to design, posing significant challenges in solving novel or complex CO problems.

Recent advances in simulated annealing (SA) (Kirkpatrick et al., 1983) and neural network (NN)-based learning (Bengio et al., 2020; Cappart et al., 2023) algorithms have redefined approaches to combinatorial optimization by minimizing dependence on manual heuristics:

- **Simulated Annealing**: SA is a general-purpose optimization algorithm that explores the solution space probabilistically, avoiding dependence on problem-specific heuristics. Although its cooling schedule and acceptance criteria require some design decisions, SA is highly adaptable across diverse problems free from detailed domain knowledge (Johnson et al., 1991).

- **Neural Network Models**: NN-based methods leverage supervised learning (Kool et al., 2019; Zhang et al., 2023; Sun & Yang, 2023; Li et al., 2023; 2024), reinforcement learning (Khalil et al., 2017; Qiu et al., 2022; Feng & Yang, 2025) or unsupervised learning (Karalias & Loukas, 2020; Wang et al., 2022; Wang & Li, 2023; Sanokowski et al., 2024) to learn optimization strategies directly from data. By automating the process, these models replace hand-crafted heuristics with learned representations and decision-making processes, enabling tailored solutions refined through training rather than manual adjustment.

Langevin dynamics (LD) (Welling & Teh, 2011) and related diffusion models (whose inference is built on LD) (Sohl-Dickstein et al., 2015; Ho et al., 2020; Song & Ermon, 2019; Song et al., 2020) have greatly advanced the recent development of SA- and NN-based solvers. The key idea of LD is to guide the iterative sampling via the gradient for a more efficient searching/generation process. This gradient-informed approach has been adapted to the discrete domain—examples include GWG (Grathwohl et al., 2021), PAFS (Sun et al., 2022), and the discrete Langevin sampler (Zhang et al., 2022)—and yields SA solvers that attain state-of-the-art (SOTA) results on various CO bench-

[1] Language Technologies Institute, Carnegie Mellon University. Correspondence to: Shengyu Feng <shengyuf@cs.cmu.edu>.

*Proceedings of the $42^{nd}$ International Conference on Machine Learning*, Vancouver, Canada. PMLR 267, 2025. Copyright 2025 by the author(s).

marks (Sun et al., 2023). Meanwhile, discrete diffusion models have also shown substantial gains. For example, DIFUSCO (Sun & Yang, 2023) adopts continuous diffusion models from computer vision to address the discrete nature of CO problems, outperforming previous end-to-end neural models in both the solution quality and computational efficiency. Additionally, DiffUCO (Sanokowski et al., 2024) generalizes DIFUSCO by eliminating the need for labeled training data, using unsupervised learning for CO problems. However, existing discrete LD/diffusion methods are mostly adapted from the methods in the continuous domain, this raises important questions: Is there any difference between continuous optimization and combinatorial optimization? Do these adapted methods sufficiently consider the nature of discrete data? Exploring these questions is the central focus of this paper.

Our key observation is that the optimization process is more prone to local optima in a discrete domain than in a continuous one. That is, local optima in a continuous domain typically have a zero gradient (under the smoothness condition), but this is often not true in a discrete domain, where the gradient may be very large in magnitude, but pointing to an infeasible region. Such a difference makes the escaping of local optima more difficult in a discrete domain than in a continuous one, with the common strategy of adding random noise as in LD. We propose to address this issue by enforcing a constant norm of the expected Hamming distance between the sampled solution and the current solution during the searching process. In other words, we control the magnitude of the update in LD, encouraging the search to explore more promising areas. We name this sampling method *Regularized Langevin Dynamics (RLD)*. We apply RLD on both SA- and NN-based CO solvers, leading to Regularized Langevin Simulated Annealing (RLSA) and Regularized Langevin Neural Network (RLNN). Our empirical evaluation on three CO problems demonstrate the significant improvement of RLSA and RLNN over both SA and NN baselines. Notably, RLSA only needs 20% runtime to outperform the previous SOTA SA baselines. And it shows a clear efficiency advantage with either less or more sampling steps. Compared with previous diffusion models, RLNN could be efficiently trained with a local optimization objective in an unsupervised manner, eliminating the need for labeled data or the estimation of a long-term return throughout the sampling process.

To summarize, we propose a new variant of discrete Langevin dynamics for combinatorial optimization by regularizing the expected update magnitude on the current solution at each step. Our method is featured by its simplicity, effectiveness, and wide applicability to both SA- and NN-based solvers, indicating its strong potential in addressing CO problems.

## 2. Related Work

### 2.1. Simulated Annealing for CO

Simulated Annealing (SA) (Metropolis et al., 1953; Hastings, 1970; Neal, 1996; IBA, 2001) is a classic metaheuristic for combinatorial optimization (WANG et al., 2009; Bhattacharya et al., 2014; Tavakkoli-Moghaddam et al., 2007; Seçkiner & Kurt, 2007; Chen & Ke, 2004). However, traditional SA methods often require up to millions of sampling steps for convergence, which limits its applicability on large-scale instances. The recent advances in discrete Markov Chain Monte Carlo (MCMC) offer the potential solutions by accelerating the convergence with locally-balanced proposals (Zanella, 2017). For example, GWG (Grathwohl et al., 2021) uses a first-order approximation to estimate energy changes within the $1$-Hamming-ball neighborhood, steering each proposal toward high-density regions. To mitigate local-optima issues caused by small neighborhoods, PAS/PAFS (Sun et al., 2022) instead samples $d$ coordinates sequentially at each step. Their SA variant iSCO (Sun et al., 2023) achieves SOTA performance on various CO benchmarks. To address the inefficiency of sequential sampling in PAS/PAFS, Zhang et al. (2022) derive a discrete Langevin sampler from the continuous domain, assigning gradient-informed change probabilities to all coordinates in parallel. However, direct transformation of the discrete Langevin sampler into an SA framework overlooks key domain differences, resulting in severe local-optima issues. In this work, we build on these discrete MCMC developments by proposing a simple method that unifies parallel sampling with effective local-optima escape.

### 2.2. Neural Solvers for CO

The neural network (NN) models have recently garnered vast attention in solving CO problems (Bengio et al., 2020; Cappart et al., 2023; Feng et al., 2025b). The NN-based solvers could be roughly categorized into three classes according to the training methods, namely the supervised learning-based (Li et al., 2018a; Gasse et al., 2019; Sun & Yang, 2023; Li et al., 2023; 2024), reinforcement learning-based (Khalil et al., 2017; Qiu et al., 2022; Feng & Yang, 2025), and unsupervised learning-based (Karalias & Loukas, 2020; Wang et al., 2022; Wang & Li, 2023; Zhang et al., 2023; Sanokowski et al., 2024). Although neural networks offer strong expressivity, the end-to-end neural solvers still struggle with the inherent non-convexity of CO problems. To address this, recent works (Sun & Yang, 2023; Li et al., 2023; 2024) have introduced discrete diffusion models, leveraging their success in capturing complex, multimodal distributions from image generation. However, these diffusion approaches depend on high-quality training data—often expensive to generate for large-scale CO instances, sometimes requiring hours per instance. DiffUCO (Sanokowski

et al., 2024) alleviates this by using unsupervised learning guided by energy-based models, but its training still relies on reinforcement learning across the entire sampling trajectory, leading to inefficiency. In contrast, our RLNN method could be optimized only through a local objective, dramatically improving training efficiency while achieving comparable performance to SOTA NN-based solvers.

## 3. Preliminary

### 3.1. Combinatorial Optimization Problem

Following Papadimitriou & Steiglitz (1982), we formulate the combinatorial optimization (CO) problem as a constrained optimization problem, i.e.,

$$\min_{\mathbf{x} \in \{0,1\}^N} a(\mathbf{x}) \quad \text{s.t.} \quad b(\mathbf{x}) = 0, \tag{1}$$

where $a(\mathbf{x})$ stands for the target to optimize and $b(\mathbf{x}) \geq 0$ corresponds to the amount of constraint violation (0 means no violation). In particular, we focus on the penalty form that can be written as

$$\min_{\mathbf{x} \in \{0,1\}^N} H(\mathbf{x}) = a(\mathbf{x}) + \beta b(\mathbf{x}), \tag{2}$$

where $\beta > 0$ is the penalty coefficient that should be sufficiently large, such that the minimum of Equation 2 corresponds to the feasible solution in Equation 1. $H(\mathbf{x})$ is also generally named as the energy function, and its associated energy-based model (EBM) is defined as

$$p_\tau(\mathbf{x}) = \frac{\exp(-H(\mathbf{x})/\tau)}{Z}, \tag{3}$$

where $\tau > 0$ is the temperature controlling the smoothness of $p_\tau(\mathbf{x})$, and $Z = \sum_{\mathbf{x} \in \{0,1\}^N} \exp(-H(\mathbf{x})/\tau)$ is the normalization factor, typically intractable. When $\tau$ is small, the probability mass of $p_\tau$ tends to concentrate around low-energy samples, making the task of solving Equation 1 equivalent to sampling from $p_\tau(\mathbf{x})$. Markov Chain Monte Carlo (MCMC) (Lecun et al., 2006) is the most widely used method for sampling from the EBM defined above. However, directly applying MCMC may lead to inefficiencies due to the non-smoothness introduced by the small $\tau$. To mitigate this issue, the simulated annealing (SA) technique is commonly employed to gradually decrease $\tau$ towards zero during the MCMC process.

### 3.2. Langevin Dynamics

Langevin dynamics (LD) (Welling & Teh, 2011) is an efficient MCMC algorithm initially developed in the continuous domain. It takes a noisy gradient ascent update at each step to gradually increase the log-likelihood of the sample:

$$\mathbf{x}' = \mathbf{x} + \frac{\alpha}{2} s(\mathbf{x}) + \sqrt{\alpha}\zeta, \quad \zeta \sim \mathcal{N}(0, \mathbf{I}_{N \times N}), \tag{4}$$

where $s(\mathbf{x}) = \nabla \log p(\mathbf{x})$ is known as the score function (gradient of the log likelihood), and $\alpha > 0$ represents the step size. By iteratively performing the above update, the sample $\mathbf{x}$ would eventually end up at a stationary distribution approximately equal to $p(\mathbf{x})$.

Recently, Zhang et al. (2022) have extended LD to a discrete Langevin sampler by rewriting Equation 4 as

$$
\begin{aligned}
q(\mathbf{x}'|\mathbf{x}) &= \frac{\exp\left(-\frac{1}{2\alpha}\|\mathbf{x}' - \mathbf{x} - \frac{\alpha}{2}s(\mathbf{x})\|_2^2\right)}{Z(\mathbf{x})} \\
&= \frac{\exp\left(\frac{1}{2}s(\mathbf{x})^T(\mathbf{x}' - \mathbf{x}) - \frac{1}{2\alpha}\|\mathbf{x}' - \mathbf{x}\|_2^2\right)}{Z(\mathbf{x})}.
\end{aligned}
\tag{5}
$$

The above distribution could be factorized coordinatewisely, i.e., $q(\mathbf{x}'|\mathbf{x}) = \prod_{i=1}^N q(\mathbf{x}'_i|\mathbf{x})$, into a set of categorical distributions:

$$q(\mathbf{x}'_i|\mathbf{x}) \propto \exp\left(\frac{1}{2}s(\mathbf{x})_i(\mathbf{x}'_i - \mathbf{x}_i) - \frac{(\mathbf{x}'_i - \mathbf{x}_i)^2}{2\alpha}\right). \tag{6}$$

When $\mathbf{x}$ is a binary vector, we can obtain the flipping (changing the value of $\mathbf{x}_i$ from 0 to 1, or 1 to 0) probability:

$$q(\mathbf{x}'_i = 1 - \mathbf{x}_i|\mathbf{x}) = \sigma\left(\frac{1}{2}s(\mathbf{x})_i(1 - 2\mathbf{x}_i) - \frac{1}{2\alpha}\right), \tag{7}$$

where $\sigma(\cdot)$ stands for the sigmoid function.

In particular, it can be shown that the discrete Langevin sampler is a first-order approximation to the locally-balanced proposal (Zanella, 2017):

$$q(\mathbf{x}'|\mathbf{x}) \propto \sqrt{p(\mathbf{x}')/p(\mathbf{x})}k(\mathbf{x}, \mathbf{x}'), \tag{8}$$

where $k(\cdot, \cdot)$ is a symmetric function corresponding to $\exp\left(-\frac{\|\mathbf{x}' - \mathbf{x}\|_2^2}{2\alpha}\right)$ here, and $\log p(\mathbf{x}') - \log p(\mathbf{x})$ (inside the exponential) is approximated by $s(\mathbf{x})^T(\mathbf{x}' - \mathbf{x})$.

## 4. Method

### 4.1. Regularized Langevin Dynamics

Although the discrete Langevin sampler can be directly converted into an SA solver by annealing the temperature $\tau$, it nonetheless frequently gets stuck in local optima (see Figure 1 in Section 5.3). As $\tau$ decreases, the gradient term dominates the transition probabilities, driving the flipping probabilities to zero and confining the next sample to a very small neighborhood of the current solution. In order to effectively avoid this undesired behavior, we propose to regularize the expected Hamming distance, i.e., the number of changed coordinates between the sampled and current solutions. Concretely, we start from a locally-balanced proposal and impose:

$$
\begin{aligned}
q(\mathbf{x}'|\mathbf{x}) &\propto \sqrt{p(\mathbf{x}')/p(\mathbf{x})}k(\mathbf{x}, \mathbf{x}'), \\
\text{s.t.} \quad & \mathbb{E}_{q(\mathbf{x}'|\mathbf{x})}[\text{Ham}(\mathbf{x}', \mathbf{x})] = d,
\end{aligned}
\tag{9}
$$

where $\mathrm{Ham}(\cdot, \cdot)$ stands for the Hamming distance and $d$ represents the regularized step size. For simplicity, we always treat $d$ as a positive integer in our design. Note that, $k(\mathbf{x}, \mathbf{x}')$ does not necessarily remain symmetric (and thus the locally-balanced condition may not be satisfied) once the regularization constraint is enforced; instead, it can be chosen to satisfy the constraint at each state.

Since the exact ratio $p(\mathbf{x}')/p(\mathbf{x})$ is typically intractable, we follow the discrete Langevin sampler to substitute its first-order Taylor series expansion:

$$q(\mathbf{x}'|\mathbf{x}) \propto \exp\left(\frac{1}{2}s(\mathbf{x})^T(\mathbf{x}' - \mathbf{x})\right)k(\mathbf{x}, \mathbf{x}'), \tag{10}$$
$$\text{s.t.} \quad \mathbb{E}_{q(\mathbf{x}'|\mathbf{x})}[\mathrm{Ham}(\mathbf{x}', \mathbf{x})] = d.$$

Empirically, this simple regularization technique substantially reduces the tendency to get stuck in local optima. We therefore call our approach *Regularized Langevin Dynamics (RLD)* and proceed to describe how it integrates into both SA- and NN-based CO solvers.

### 4.2. Regularized Langevin Simulated Annealing

When $\mathbf{x}$ is binary, we could follow Zhang et al. (2022) to let $k(\mathbf{x}, \mathbf{x}') = \exp\left(-\frac{\|\mathbf{x}' - \mathbf{x}\|_2^2}{2\alpha}\right)$ and explicitly write out the expectation in Equation 10 with the flipping probabilities:

$$\sum_{i=1}^{N} \sigma\left(\frac{1}{2}s(\mathbf{x})_i(1 - 2\mathbf{x}_i) - \frac{1}{2\alpha}\right) = d. \tag{11}$$

Since the gradient of the energy function could be computed in a closed form for various CO problems, here we first assume $\nabla H(\mathbf{x})$ is available. Note that the score function of the EBM could be written as

$$s_\tau(\mathbf{x}) = \log p_\tau(\mathbf{x}) = -\frac{1}{\tau}\nabla H(\mathbf{x}). \tag{12}$$

To avoid clutter, we denote $\Delta = (2\mathbf{x} - 1) \odot \nabla H(\mathbf{x})$, whose $i$-th coordinate approximates the drop of the energy function if we flip the value of $\mathbf{x}_i$.

The exact solving of Equation 11 is challenging due to the presence of the sigmoid function. However, when $\tau \to 0$, we observe that the sigmoid function is approximately an indicator function:

$$\lim_{\tau \to 0} \sigma\left(\frac{1}{2\tau}\Delta_i - \frac{1}{2\alpha}\right) = \mathbb{1}\left(\frac{1}{2\tau}\Delta_i - \frac{1}{2\alpha} > 0\right). \tag{13}$$

This property allows us to efficiently regularize the SA algorithm with the $d$-th largest element in $\Delta$, denoted as $\Delta_{(d)}$. We then obtain the flipping probabilities by letting $\frac{1}{\alpha} = \frac{\Delta_{(d)} + \epsilon}{\tau}$, where $\epsilon \geq 0$ (e.g., $10^{-6}$) is optional:

$$q(\mathbf{x}_i' = 1 - \mathbf{x}_i|\mathbf{x}) = \sigma\left(\frac{1}{2\tau}(\Delta_i - \Delta_{(d)} - \epsilon)\right). \tag{14}$$

We call the resultant SA algorithm as *Regularized Langevin Simulated Annealing (RLSA)*, whose details are summarized in Algorithm 1.

---

**Algorithm 1** Regularized Langevin Simulated Annealing

1: **Input**: $T$, $d$ and $\tau_0$
2: Initialize $\mathbf{x} \in \{0, 1\}^N$; $\mathbf{x}^* \leftarrow \mathbf{x}$
3: **for** $t = 1, \cdots, T$ **do**
4: $\quad \tau \leftarrow \tau_0\left(1 - \frac{t-1}{T}\right)$
5: $\quad \Delta \leftarrow (2\mathbf{x} - 1) \odot \nabla H(\mathbf{x})$
6: $\quad$ **for** $i = 1, \cdots, N$ **do**
7: $\quad\quad p \leftarrow \sigma\left(\frac{1}{2\tau}(\Delta_i - \Delta_{(d)} - \epsilon)\right)$
8: $\quad\quad c \sim \texttt{Bernoulli}(p)$
9: $\quad\quad \mathbf{x}_i \leftarrow \mathbf{x}_i(1 - c) + (1 - \mathbf{x}_i)c$
10: $\quad$ **end for**
11: $\quad$ **if** $H(\mathbf{x}) < H(\mathbf{x}^*)$ **then**
12: $\quad\quad \mathbf{x}^* \leftarrow \mathbf{x}$
13: $\quad$ **end if**
14: **end for**
15: **return** $\mathbf{x}^*$

---

It is worth noting that $k(\mathbf{x}, \mathbf{x}') = \exp\left(-\frac{\|\mathbf{x}' - \mathbf{x}\|_2^2}{2\alpha}\right)$ is chosen here just for simplicity, but other strategies may perform just as well. For instance, one could omit the $-\frac{1}{\alpha}$ term and instead normalize the sigmoid outputs:

$$\tilde{p}_i = \frac{\sigma\left(\frac{1}{2}s(\mathbf{x})_i(1 - 2\mathbf{x}_i)\right)}{\sum_{j=1}^{N}\sigma\left(\frac{1}{2}s(\mathbf{x})_j(1 - 2\mathbf{x}_j)\right)}d, \tag{15}$$

$$p_i = \min\{1, \max\{0, \tilde{p}_i\}\}, \tag{16}$$

so that $\sum_i p_i \approx d$. Investigating alternative kernel functions or normalization strategies—potentially adapted to specific problem structures—offers a promising avenue for future work aimed at further minimizing the approximation error.

In our implementation, the elementwise sampling is run in parallel and we maintain $K$ independent SA processes simultaneously. The whole algorithm could be implemented in a few lines and accelerated with GPU-based deep learning frameworks, such as PyTorch (Paszke et al., 2017) and Jax (Bradbury et al., 2018). An example PyTorch code is attached in Appendix C.

Given the overall RLSA framework, we now address the question of how to compute the gradient of the energy function. Numerous CO problems are defined on graphs and could be formulated in the quadratic form, known as QUBO (Lucas, 2014). Let $\mathcal{G} = (\mathcal{V}, \mathcal{E})$ be an undirected graph, with node set $\mathcal{V} = \{1, \cdots, N\}$, edge set $\mathcal{E} \in \mathcal{V} \times \mathcal{V}$, and adjacency matrix $\mathbf{A} \in \{0, 1\}^{N \times N}$. In this work, we focus on the following three problems, which have been commonly used in benchmark evaluations for CO solvers.

**Maximum Independent Set.** The maximum independent set (MIS) problem aims to select the largest subset of nodes of the graph $\mathcal{G}$, without any adjacent pair. Denote a selected node as $\mathbf{x}_i = 1$ and an unselected one as $\mathbf{x}_i = 0$, the energy function of MIS could be expressed as

$$
\begin{aligned}
H(\mathbf{x}) &= -\sum_{i=1}^{N} \mathbf{x}_i + \beta \sum_{(i,j)\in\mathcal{E}} \mathbf{x}_i\mathbf{x}_j \\
&= -\mathbf{1}^{\top}\mathbf{x} + \beta\frac{\mathbf{x}^{\top}\mathbf{A}\mathbf{x}}{2},
\end{aligned}
\tag{17}
$$

As this is a quadratic function, it is straightforward to compute the gradient of the energy function as

$$
\nabla H(\mathbf{x}) = -\mathbf{1} + \beta\mathbf{A}\mathbf{x}. \tag{18}
$$

**Maximum Clique.** The maximum clique (MCl) stands for the largest subset of nodes in a graph such that every two nodes in the set are adjacent to each other. It could actually be expressed as the MIS problem in the complete graph, with the energy function:

$$
H(\mathbf{x}) = -\sum_{i=1}^{N} \mathbf{x}_i + \beta \sum_{(i,j)\notin\mathcal{E}} \mathbf{x}_i\mathbf{x}_j. \tag{19}
$$

In order to represent the energy function with the adjacency matrix $\mathbf{A}$, we can write the penalty as $\left(\left(\sum_{i=1}^{N} \mathbf{x}_i\right)^2 - \sum_{i=1}^{N} \mathbf{x}_i^2\right)/2 - \sum_{(i,j)\in\mathcal{E}} \mathbf{x}_i\mathbf{x}_j$, reformulating the energy function and its gradient as

$$
H(\mathbf{x}) = -\mathbf{1}^{\top}\mathbf{x} + \beta\frac{(\mathbf{1}^{\top}\mathbf{x})^2 - \mathbf{x}^{\top}\mathbf{x} - \mathbf{x}^{\top}\mathbf{A}\mathbf{x}}{2}, \tag{20}
$$

$$
\nabla H(\mathbf{x}) = -\mathbf{1} + \beta\big((\mathbf{1}^{\top}\mathbf{x})\mathbf{1} - \mathbf{x} - \mathbf{A}\mathbf{x}\big). \tag{21}
$$

**Maximum Cut.** The maximum cut (MCut) problem looks to partition the nodes into two sets so that the number of edges between two sets is maximized. Here we use $\mathbf{x}_i = 1$ and $\mathbf{x}_i = 0$ to represent the belonging to two sets, and the energy function could be expressed as

$$
\begin{aligned}
H(\mathbf{x}) &= -\sum_{(i,j)\in\mathcal{E}} \frac{1 - (2\mathbf{x}_i - 1)(2\mathbf{x}_j - 1)}{2} \\
&= \mathbf{x}^{\top}\mathbf{A}\mathbf{x} - \mathbf{1}^{\top}\mathbf{A}\mathbf{x},
\end{aligned}
\tag{22}
$$

whose gradient could be accordingly computed as

$$
\nabla H(\mathbf{x}) = \mathbf{A}(2\mathbf{x} - \mathbf{1}). \tag{23}
$$

### 4.3. Regularized Langevin Neural Network

The efficacy of gradient-guided SA solvers, including RLSA, is critically dependent on the knowledge of $\nabla H(\mathbf{x})$

(Sun et al., 2022). However, $\nabla H(\mathbf{x})$ often lacks a closed-form expression or is too costly to compute directly, necessitating the gradient approximation. To evaluate RLD under these conditions, we employ a neural network to approximate the sampling distribution $q_\theta(\mathbf{x}'|\mathbf{x})$, thereby demonstrating its effectiveness even when only an approximate gradient is available.

Here we still utilize a mean-field decomposition, letting $q_\theta(\mathbf{x}'|\mathbf{x}) = \Pi_{i=1}^{N} q_\theta(\mathbf{x}'_i|\mathbf{x})$. The RLD update in Equation 10 could be translated into the following training loss

$$
\begin{aligned}
l_{RLD}(\theta; \mathbf{x}, d, \lambda) &= \mathbb{E}_{q_\theta(\mathbf{x}'|\mathbf{x})}[H(\mathbf{x}')] \\
&+ \lambda\bigg(\sum_{i}^{N} q_\theta(\mathbf{x}'_i = 1 - \mathbf{x}_i|\mathbf{x}) - d\bigg)^2.
\end{aligned}
\tag{24}
$$

The first term minimizes the conditional expectation of the energy function after the one-step update. When the expectation is tractable, this term is equivalent to the unsupervised learning loss of Erdőes Goes Neural (EGN) (Karalias & Loukas, 2020); otherwise, we could optimize this loss by estimating the policy gradient, i.e., $\mathbb{E}_{q_\theta(\mathbf{x}'|\mathbf{x})}[H(\mathbf{x}')\nabla\log q_\theta(\mathbf{x}'|\mathbf{x})]$, via Monte Carlo methods. In this work, we focus on verifying the effectiveness of RLD and adopts the unsupervised training loss for simplicity. The second term regularizes the expected Hamming distance between the two solutions, with $\lambda$ being the regularization coefficient. We name this NN-based solver as *Regularized Langevin Neural Network (RLNN)*.

We train RLNN in a similar fashion to reinforcement learning through sampling and update, but without the need to account the future states except the immediate next one. This allows RLNN to circumvent the high variance in estimating the future return when trained with a long sampling process. In detail, each time we sequentially sample $T'$ samples with the current proposal distribution $q_\theta(\mathbf{x}'|\mathbf{x})$, then for each sample, we train RLNN to minimize the loss in Equation 24. The training algorithm of RLNN is summarized in Algorithm 2.

---

**Algorithm 2** Regularized Langevin Neural Network

1: **Input**: $T'$, $d$, $\lambda$
2: Initialize $\theta$
3: **while** the stopping criterion is not met **do**
4:      Initialize $\mathbf{x} \in \{0,1\}^N$, $\mathcal{D} = \{\mathbf{x}\}$
5:      **for** $t = 1, \cdots, T'$ **do**
6:          $\mathbf{x}' \sim q_\theta(\mathbf{x}'|\mathbf{x})$
7:          $\mathcal{D} \leftarrow \mathcal{D} \cup \{\mathbf{x}'\}$
8:          $\mathbf{x} \leftarrow \mathbf{x}'$
9:      **end for**
10:      $\theta \leftarrow \min_\theta \mathbb{E}_{\mathbf{x}\in\mathcal{D}}[l_{RLD}(\theta; \mathbf{x}, d, \lambda)]$
11: **end while**
12: **return** $\theta$

---

Similarly, we maintain $K'$ parallel sampling processes in our implementation to obtain more efficient training data collection. During the inference time, we simply sample from $q_\theta(\mathbf{x}'|\mathbf{x})$ sequentially for $T$ steps with $K$ processes run in parallel. Note that temperature annealing is not used here as we do not find it useful and we simply leave $\tau = 1$.

### 4.4. Connection to Normalized Gradient Descent

Our proposed RLD method is closely related to the normalized gradient descent (NGD) method (Cortés, 2006) in the continuous domain:

$$\mathbf{x}' = \mathbf{x} - \alpha \frac{\nabla f(\mathbf{x})}{\|\nabla f(\mathbf{x})\|_2}. \qquad (25)$$

NGD is developed to address the vanishing/exploding gradient by normalizing the L2 norm of the gradient for a scale-invariant update at each step. Our method, especially RLSA, could be treated as a discrete version of this approach by regularizing the Hamming distance between the solutions before and after the update. And both methods accelerate the descent under the smoothness condition.

The key difference between the two lies in the case when $\Delta_{(d)} < 0$, RLSA could not be translated into a gradient descent algorithm to minimize the energy function, since $\alpha = \frac{\tau}{\Delta_{(d)}} < 0$ (ignore $\epsilon$) reverses the direction of gradient descent. Instead, RLSA should be treated now as a way to escape local optima without dramatically increasing the energy function. Utilizing Equation 5, we can fully express the proposal at this state as

$$q(\mathbf{x}'|\mathbf{x}) = \frac{\exp\left(-\frac{\Delta_{(d)}}{2\tau}\|\mathbf{x}' - \mathbf{x} + \frac{1}{2\Delta_{(d)}}\nabla H(\mathbf{x})\|_2^2\right)}{Z(\mathbf{x})}. \qquad (26)$$

It should be noted that the density of $q(\mathbf{x}'|\mathbf{x})$ increases with respect to the distance from $\mathbf{x} - \frac{1}{2\Delta_{(d)}}\nabla H(\mathbf{x})$, which is the **gradient ascent direction** (note that $\Delta_{(d)}$ is negative here) of the energy function. This behavior is desirable due to the different property of the local optima in the discrete data, which may not vanish to zero but point to an infeasible region (i.e., with $\Delta$ negative in all coordiantes).

Let us take MIS as an example, whose local optimum corresponds to a maximal independent set, that is, each unselected node has at least one neighbor in the set. At the local optimum, the gradient at the selected node $\mathbf{x}_i$ is $\nabla H(\mathbf{x})_i = -1$, which points to the direction of increasing $\mathbf{x}_i$, and is infeasible since $\mathbf{x}_i \leq 1$. Similarly, the gradient at an unselected node is bounded by $\nabla H(\mathbf{x})_i \geq -1 + \beta > 0$, which points to the direction of decreasing the value, and is also infeasible. Since the gradient descent direction is not informative, RLSA would try to escape this local optima but avoid the steepest direction to increase the energy function, i.e., the gradient ascent direction $\mathbf{x} - \frac{1}{2\Delta_{(d)}}\nabla H(\mathbf{x})$.

With the same example, we could also see why the standard discrete Langevin sampler (Zhang et al., 2022) with a constant step size fails here. Since LD is a first-order approximation of the locally-balanced proposal (Zanella, 2017), a small $\alpha$ is needed to make the approximation accurate. However, a small $\alpha$ would also lead to a strong penalization on the magnitude of the update. At local optima, $\Delta_i < 0$ would further discourage the change, as indicated in Equation 7. Therefore, additional efforts are needed to help LD escape the local optima beyond the force of random noise. This distinction between combinatorial and continuous optimization highlights the significance of RLD.

## 5. Experiments

### 5.1. Experimental Setup

**Benchmark datasets.** Following Zhang et al. (2023), we evaluate MIS and MCl using Revised Model B (RB) graphs (Xu & Li, 2000), and we evaluate MCut using Barabási-Albert (BA) graphs (Barabási & Albert, 1999). Following Qiu et al. (2022), we also include Erdős-Rényi (ER) graphs for MIS. As in prior work, each graph type is generated at two scales:

- **RB and BA graphs**: small (200–300 nodes) and large (800–1200 nodes).

- **ER graphs**: small (700–800 nodes) and large (9000–11000 nodes).

The large-scale ER graphs serve as a *transfer-testing set* for models trained on small-scale ER. We append a suffix "-[$n$–$N$]" to each graph name to indicate its size range.

For RB and BA (both scales) and ER-[700–800], we use 1000 graphs for training, 100 for validation, and 500 (RB/BA) or 128 (ER-[700–800]) for testing. ER-[9000–11000] is reserved solely for testing (16 graphs).

**Baselines.** Following Qiu et al. (2022) and Zhang et al. (2023), we categorize our baselines as the classical operations research solvers (OR), (human-designed) heuristic solvers (H), supervised learning-based solvers (SL), reinforcement learning-based solvers (RL), and unsupervised learning-based solvers (UL). For MIS, we have the integer linear programming solver Gurobi (Gurobi Optimization, LLC, 2023) and MIS-specific solver KaMIS (Großmann et al., 2023) as the OR baselines, and the recent SA method iSCO (Sun et al., 2023) as a heuristic baseline. In the SL category, we include INTEL (Li et al., 2018b), DGL (Böther et al., 2022), and DIFUSCO (Sun & Yang, 2023). In the RL category, we have PPO (Ahn et al., 2020) and DIMES (Qiu et al., 2022). In the UL category, we use LTFT (Zhang et al., 2023) and DiffUCO (Sanokowski et al., 2024). For

*Table 1.* Comparative results on the *Mximum independent Set (MIS)* problem. On each dataset, we bold the best result and color the second-best one in green. By "best" or "second best", we exclude the OR solvers (Gurobi and KaMIS) as their runtime is excessively large, preventing a fair comparison with the methods in other categories.

| MIS | | RB-[200–300] | | RB-[800–1200] | | ER-[700–800] | | ER-[9000–11000] | |
|---|---|---|---|---|---|---|---|---|---|
| METHOD | TYPE | SIZE ↑ | TIME ↓ | SIZE ↑ | TIME ↓ | SIZE ↑ | TIME ↓ | SIZE ↑ | TIME ↓ |
| Gurobi | OR | 19.98 | 47.57m | 40.90 | 2.17h | 41.38 | 50.00m | — | — |
| KaMIS | OR | 20.10 | 1.40h | 43.15 | 2.05h | 44.87 | 52.13m | 381.31 | 7.60h |
| PPO | RL | 19.01 | 1.28m | 32.32 | 7.55m | — | — | — | — |
| INTEL | SL | 18.47 | 13.07m | 34.47 | 20.28m | 34.86 | 6.06m | 284.63 | 5.02m |
| DGL | SL | 17.36 | 12.78m | 34.50 | 23.90m | 37.26 | 22.71m | — | — |
| DIMES | RL | — | — | — | — | 42.06 | 12.01m | 332.80 | 12.72m |
| DIFUSCO | SL | 18.52 | 16.05m | — | — | 41.12 | 26.67m | — | — |
| LTFT | UL | 19.18 | 32s | 37.48 | 4.37m | — | — | — | — |
| DiffUCO | UL | 19.24 | 54s | 38.87 | 4.95m | — | — | — | — |
| iSCO | H | 19.29 | 2.71m | 36.96 | 11.26m | 42.18 | 1.45m | 365.37 | 1.10h |
| RLNN | UL | 19.52 | 1.64m | 38.46 | 6.24m | 43.34 | 1.37m | 363.34 | 11.76m |
| RLSA | H | **19.97** | 35s | **40.19** | 1.85m | **44.10** | 20s | **375.31** | 1.66m |

*Table 2.* Comparative results on the *Maximum Clique (MCl)* and *Maximum Cut (MCut)* problems. On each dataset, we bold the best result and color the second-best one in green. By "best" or "second best", we exclude the OR solvers (Gurobi and SDP) as their runtime is excessively large, preventing a fair comparison with the methods in other categories.

| MCl | | RB-[200–300] | | RB-[800–1200] | | MCut | | BA-[200–300] | | BA-[800–1200] | |
|---|---|---|---|---|---|---|---|---|---|---|---|
| METHOD | TYPE | SIZE ↑ | TIME ↓ | SIZE ↑ | TIME ↓ | METHOD | TYPE | SIZE ↑ | TIME ↓ | SIZE ↑ | TIME ↓ |
| Gurobi | OR | 19.05 | 1m55s | 33.89 | 19.67m | Gurobi | OR | 730.87 | 8.50m | 2944.38 | 1.28h |
| SDP | OR | — | — | — | — | SDP | OR | 700.36 | 35.78m | 2786.00 | 10.00h |
| Greedy | H | 13.53 | 25s | 26.71 | 25s | Greedy | H | 688.31 | 13s | 2786.00 | 3.12m |
| MFA | H | 14.82 | 27s | 27.94 | 2.32m | MFA | H | 704.03 | 1.60m | 2833.86 | 7.27m |
| EGN | UL | 12.02 | 41s | 25.43 | 2.27m | EGN | UL | 693.45 | 46s | 2870.34 | 2.82m |
| LTFT | UL | 16.24 | 42s | 31.42 | 4.83m | LTFT | UL | 704.30 | 2.95m | 2864.61 | 21.33m |
| DiffUCO | UL | 16.22 | 1.00m | — | — | DiffUCO | UL | 727.32 | 1.00m | 2947.53 | 3.78m |
| iSCO | H | 18.96 | 54s | 40.35 | 11.37m | iSCO | H | 728.24 | 1.67m | 2919.97 | 4.18m |
| RLNN | UL | 18.13 | 1.36m | 35.23 | 7.83m | RLNN | UL | 729.00 | 1.58m | 2907.18 | 3.67m |
| RLSA | H | **18.97** | 23s | **40.53** | 1.27m | RLSA | H | **733.54** | 27s | **2955.81** | 1.45m |

the non-MIS problems, the baselines include two OR methods, which are Gurobi and a semi-definite programming method (SDP) for MCut; three heuristic methods, which are greedy, mean-field annealing (MFA) (Bilbro et al., 1988) and iSCO (Sun et al., 2023); and three UL methods, which are EGN (Karalias & Loukas, 2020), LTFT (Zhang et al., 2023) and DiffUCO (Sanokowski et al., 2024), respectively. For most of these methods, we report their published results (Qiu et al., 2022; Zhang et al., 2023; Sun & Yang, 2023; Sanokowski et al., 2024; Li et al., 2024). If an NN solver has multiple variants, we only compare with the variant with the longest runtime (typically corresponding to the best result). We rerun the official code of iSCO[1] on our datasets due to the inconsistency of time measurement in this work, where the average runtime of iSCO is compared to the total runtime of its baselines. For a fair comparison, we run iSCO with the same number of steps and trials as we do for RLSA.

---

[1] https://github.com/google-research/discs

**Implementation Details.** Our implementation is based on PyTorch Geometric (Fey & Lenssen, 2019) and Accelerate (Gugger et al., 2022). We use two servers for the RLNN training, one with 8 NVIDIA RTX A6000 GPUs and the other with 10 NVIDIA RTX 2080 Ti GPUs. We evaluate all methods on the first server, using a single A6000 GPU. We find the efficiency of RLNN highly susceptible to the inductive bias of the NN architecture, e.g., a two-parameter linear model is enough to fit the gradient of MIS in Equation 19. Since the main focus of our evaluation is to verify the effectiveness of RLD under the approximate gradient, we keep the model architecture of RLNN basically the same as the ones used in prior works (Qiu et al., 2022; Sanokowski et al., 2024). Future work may further optimize the neural architecture by injecting more prior knowledge about the problem structure, as in Yau et al. (2024). In our experiment, we parameterize RLNN with a five-layer GCN (Kipf & Welling, 2017) with 128 hidden dimensions. Due to the

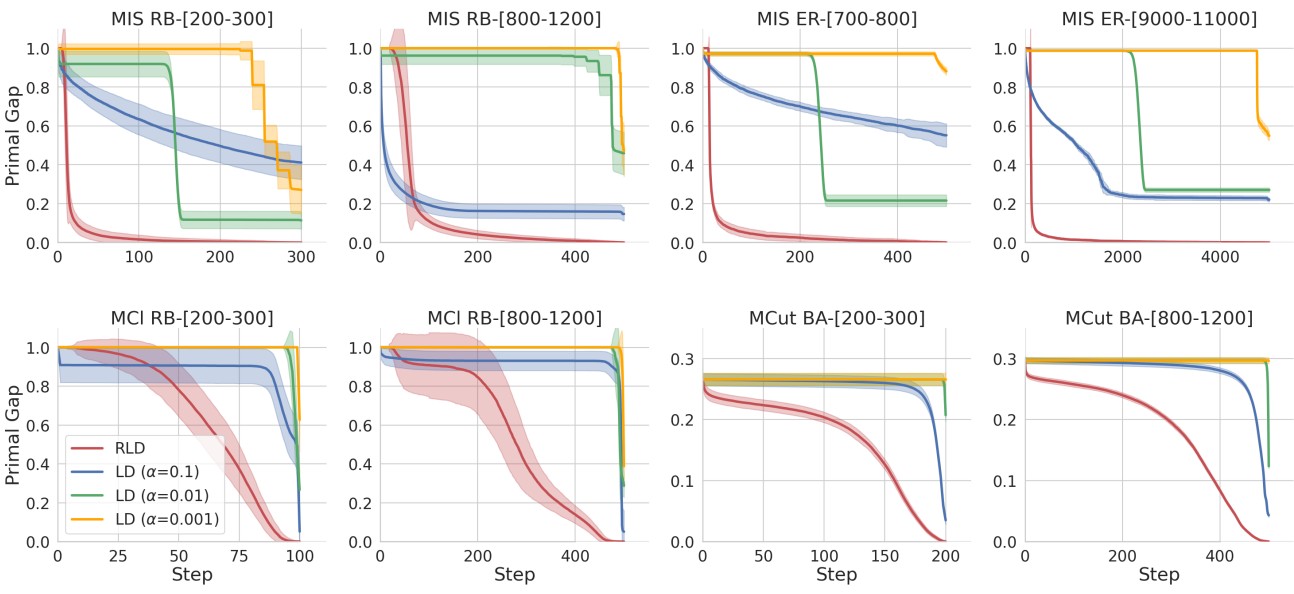

*Figure 1.* Primal-gap trajectories for SA solvers using RLD versus the standard discrete LD method (Zhang et al., 2022). The RLD (corresponding to RLSA) curve is shown in red, and the remaining curves (in distinct colors) correspond to discrete LD with various step-size settings. Solid lines denote the mean primal gap over the test set, and shaded regions represent the standard deviation.

increasing computational complexity at each step, we also accordingly reduce the number of sampling steps and trials of RLNN compared to RLSA, with other hyperparameters kept the same. We include more details in Appendix A.

### 5.2. Main Results

In performance evaluation, we compare the mean value of the achieved problem-specific objective (larger is better) of each method on each problem, including the set size for MIS, clique size for MCl and cut size for MCut. In addition, we compare the *total runtime* (lower is better) of each method throughout the test set by sequentially evaluating each instance. Since OR solvers are guaranteed to find the optimal solution with enough runtime, we do not include them for comparison.

Table 1 reports our results on the MIS problem. RLSA achieves significant improvements over SOTA NN-based methods on both RB and ER graphs, with similar or even shorter runtime. Moreover, RLSA consistently outperforms iSCO, another gradient-guided SA method, with the same number of sampling steps and independent trials. Due to its parallel sampling design, RLSA requires only 5-20% of iSCO's runtime while delivering better objective values.

Based on less prior knowledge (using an approximate gradient), RLNN shows a lower performance than RLSA, but still claims the second-best result in two of the four datasets. On ER-[9000–11000] graphs, RLNN matches iSCO's performance but uses under 20% of the runtime. Compared

to the SOTA NN baselines, RLNN shows a performance comparable to DiffUCO on the RB graphs and clearly outperforms DIMES on ER. Moreover, it should be pointed out that DiffUCO can require up to two days of training even on small graphs, e.g., RB-[200–300], whereas RLNN completes the training in under three hours regardless the graph scales, while delivering comparable performance. This highlights RLNN's superior training efficiency against previous diffusion models, thanks to its local training objective.

Table 2 summarizes our comparative results on MCl and MCut. RLSA retains a clear efficiency advantage over iSCO. Although iSCO matches RLSA's performance on MCl at both scales, RLSA and RLNN still outperform all other baselines. On MCut, RLSA consistently leads; while RLNN, DiffUCO, and iSCO achieve comparable results—all significantly better than the remaining methods.

We also include extended comparisons between iSCO and RLSA with ten times more sampling steps in Appendix B, which confirms the above findings. Overall, both RLSA and RLNN remain highly competitive across our benchmarks, with RLSA delivering impressive results on every dataset at minimal computational cost.

### 5.3. Ablation Study

To verify the effectiveness of the proposed regularization in RLD, we conduct the ablation study on RLSA and RLNN, respectively. We first compare RLD with the standard discrete LD (Zhang et al., 2022) for SA, searching the step

*Table 3.* Ablation study on the effectiveness of regularization in RLNN. The numbers correspond to the set size (larger is better).

| | **MIS** | | **MCl** | **MCut** |
|---|---|---|---|---|
| METHOD | RB-[200–300] | ER-[700–800] | RB-[200–300] | BA-[200–300] |
| RLNN w/o regularization | 18.64 | 37.73 | 16.62 | **730.20** |
| RLNN w/ regularization | **19.52** | **43.34** | **18.13** | 729.00 |

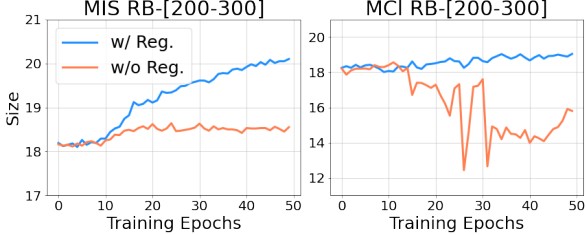

*Figure 2.* Training curves of RLNN with or without regularization. Validation performance (set/clique size) is shown.

size $\alpha$ over the set $\{0.1, 0.01, 0.001\}$. All other hyperparameters are kept the same as in RLSA. Figure 1 compares the dynamics of the primal gap (Berthold, 2014) across the sampling process. Here, the primal gap on each instance is defined as

$$\begin{cases} \frac{|H(\mathbf{x})-H(\mathbf{x}^*)|}{\max\{|H(\mathbf{x})|,|H(\mathbf{x}^*)|\}}, & \text{if } H(\mathbf{x})H(\mathbf{x}^*) \geq 0; \\ 1, & \text{otherwise,} \end{cases} \quad (27)$$

where $\mathbf{x}$ corresponds to the best solution found so far and $\mathbf{x}^*$ is a pre-computed optimal (or best known) solution.

It is evident that the standard discrete LD always ends up at a sub-optimal solution except on MCl. The searching could easily get stuck in a local optimum, indicated by the flat stage. In contrast, RLSA typically converges in fewer than 100 steps without becoming trapped. Note that our search set already includes the most common gradient-descent step sizes used for continuous problems, and even a smaller step size (e.g., 0.001) yields worse results. Unlike continuous optimization, combinatorial optimization presents unique challenges; RLD is specifically designed to address these.

We next perform an ablation study on the regularization term in Equation 24 for RLNN training. Specifically, we train RLNN on small-scale graphs both with and without regularization, and report results in Table 3. Adding regularization significantly improves RLNN's performance in most cases, with the exception of MCut—likely because MCut is unconstrained and is less prone to local optima. On all other benchmarks, training RLNN without regularization proves almost ineffective. Figure 2 visualizes this effect by plotting the set/clique size on the validation set (larger is better) for MIS and MCl.

From Figure 2, the curve without regularization (orange)

remains flat on MIS or even deteriorates after more training epochs on MCl. This highlights that only using the unsupervised loss of EGN makes sample diversity a critical concern. By contrast, the regularization term encourages RLNN to gather more diverse training samples and explore more effectively during inference, resulting in the stable performance gains shown by the blue curve throughout training.

In fact, the idea of RLNN also parallels well-known reinforcement learning techniques to encourage exploration, such as curiosity-driven exploration (Pathak et al., 2017) and soft policies (Haarnoja et al., 2017; 2018).

## 6. Conclusion & Limitation

In this work, we point out the specific challenge from the local optima in combinatorial optimization (CO), and propose a novel sampling framework called Regularized Langevin Dynamics (RLD) to tackle the issue in CO. On top of that, we develop two CO solvers, one based on simulated annealing (SA), and the other one based on neural networks. Our empirical evaluation on three classic CO problems demonstrate that our proposed methods can achieve state-of-the-art (SOTA) or near-SOTA performance with high efficiency. In particular, our proposed SA method consistently outperforms the previous SA baseline using only 20% running time, while RLNN significantly improves the training efficiency of previous diffusion models. In summary, RLD is a simple yet effective framework that shows great potential in addressing CO problems.

In this work, we only consider binary data for ease of analysis. Although the whole framework could be generalized, its effectiveness remains unclear in other CO problems with categorical, integer, or mixed integer variables. Future work may also extend it to other CO problems with global constraints, such as the Traveling Salesman Problem. Besides, we have only given an intuitive explanation of RLD in this work, but the theoretical understanding of RLD is generally missing. We also expect to address this part in the future.

## Impact Statement

This paper presents work whose goal is to advance the field of Machine Learning. There are many potential societal consequences of our work, none which we feel must be specifically highlighted here.

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

# A. Additional Experiment Details

## A.1. Overview of Hyperparameters

Tables 4 and 5 summarize the hyperparameters for RLSA and RLNN, respectively. We use *random search* to determine the initial temperature $\tau_0$ within $[0.001, 10]$ and the regularized step size $d$ within $[2, 100]$. Since larger values of $K$ and $T$ generally yield better performance, we chose $K$ and $T$ for RLSA so that its runtime matches that of the fastest baseline on each dataset. For RLNN, we allow a larger time budget by selecting $K$ and $T$ such that its runtime is comparable to most baselines. The inference is performed using the *float16* data type to accelerate tensor operations.

*Table 4.* Hyperparameters used by RLSA on all datasets.

| Problem | Dataset | $\tau_0$ | $d$ | $K$ | $T$ | $\beta$ |
|---|---|---|---|---|---|---|
| MIS | RB-[200-300] | 0.01 | 5 | 200 | 300 | 1.02 |
| | RB-[800-1200] | 0.01 | 5 | 200 | 500 | 1.02 |
| | ER-[700-800] | 0.01 | 20 | 200 | 500 | 1.001 |
| | ER-[9000-1100] | 0.01 | 20 | 200 | 5000 | 1.001 |
| MCl | RB-[200-300] | 4 | 2 | 200 | 100 | 1.02 |
| | RB-[800-1200] | 4 | 2 | 200 | 500 | 1.02 |
| MCut | BA-[200-300] | 5 | 20 | 200 | 200 | 1.02 |
| | BA-[800-1200] | 5 | 20 | 200 | 500 | 1.02 |

*Table 5.* Hyperparameters used by RLNN on all datasets.

| Problem | Dataset | $\tau_0$ | $d$ | $K$ | $T$ | $\beta$ | $K'$ | $T'$ | $\lambda$ |
|---|---|---|---|---|---|---|---|---|---|
| MIS | RB-[200-300] | 1 | 5 | 20 | 100 | 1.02 | 10 | 50 | 0.5 |
| | RB-[800-1200] | 1 | 5 | 20 | 200 | 1.02 | 10 | 300 | 0.5 |
| | ER-[700-800] | 1 | 20 | 20 | 200 | 1.001 | 10 | 500 | 0.5 |
| | ER-[9000-1100] | 1 | 20 | 20 | 800 | 1.001 | — | — | — |
| MCl | RB-[200-300] | 1 | 2 | 20 | 100 | 1.02 | 10 | 100 | 0.5 |
| | RB-[800-1200] | 1 | 2 | 20 | 200 | 1.02 | 10 | 300 | 0.5 |
| MCut | BA-[200-300] | 1 | 20 | 20 | 100 | 1.02 | 10 | 50 | 0.5 |
| | BA-[800-1200] | 1 | 20 | 20 | 200 | 1.02 | 10 | 300 | 0.5 |

## A.2. Implementation of RLNN

RLNN is parameterized by a five-layer GCN (Kipf & Welling, 2017) with 128 hidden dimensions. A linear layer is first used to project the input $\mathbf{x}$ to a 128-dim embedding $\mathbf{H}^0$. Each layer of GCN performs the following update:

$$\mathbf{H}^{l+1} = ReLU(\mathbf{U}^l \mathbf{H}^l + \mathbf{V}^l \mathbf{D}^{-1/2} \hat{\mathbf{A}} \mathbf{D}^{-1/2} \mathbf{H}^l) + \mathbf{H}^l, \tag{28}$$

where $\mathbf{U}^l$ and $\mathbf{V}^l$ are the model parameters at $l$-th layer, $\hat{\mathbf{A}} = \mathbf{A} + \mathbf{I}_{N \times N}$ is the adjacency matrix with the self loop, $\mathbf{D}$ is a diagonal degree matrix with $\mathbf{D}_{ii} = \sum_{j=1}^N \hat{\mathbf{A}}_{ij}$. The output hidden representation is projected to a scalar via a linear layer, and then a sigmoid activation yields the flipping probability $q_\theta(\mathbf{x}'_i = 1 - \mathbf{x}_i | \mathbf{x})$ for each node.

We train RLNN with 50 epochs on all datasets, except RB-[700–800] for MCl, where we use 80 epochs because the model does not converge in 50 epochs. On each graph, we sample $K'$ trajectories of length $T'$, yielding $K'T'$ training samples. The batch size is set to 32 per GPU, and we optimize with Adam at a learning rate of 0.0001.

In terms of training time, our model completes training on small-scale graphs (except ER-[700–800]) in under half an hour using eight RTX A6000 GPUs, and the larger instances finish within one hour; on a server with ten RTX 2080 Ti GPUs, runtimes are slightly longer, but the longest experiment still finishes within three hours. By comparison, baseline methods

such as DiffUCO ([Sanokowski et al., 2024](#)) require two days to train a single model on RB–[200–300]. We attribute this efficiency advantage to the local training objective of RLNN, which avoids estimating long-term high-variance reward signals with standard reinforcement learning methods, as employed by DiffUCO.

Note that the *float32* data type is used during RLNN training, which is switched to *float16* at the inference time.

### A.3. Postprocessing

We postprocess the lowest-energy solutions to enforce feasibility, though in practice our methods almost always produce valid solutions, so we use a simple greedy decoder.

Specifically, we calculate the gradient of the current solution $\mathbf{x}$: if $\max \Delta_i < 0$, then the solution has already reached a local optimum and the feasibility is guaranteed; otherwise, we flip the coordinate corresponding to $\arg\max \Delta_i$. This process is iterated until the convergence.

## B. Comparison under Longer Runtime

To assess whether RLSA retains its advantage when given longer runtime, we run both RLSA and iSCO for ten times more iterations than those specified in Table 4. The results in Tables 6 and 7 show that, although iSCO matches RLSA on some small-scale instances, RLSA still outperforms on large-scale datasets while requiring substantially less computation time.

Furthermore, the results demonstrate that RLSA achieves performance comparable to exact solvers across multiple benchmarks—especially on large-scale problem instances—underscoring its effectiveness in CO.

*Table 6.* Comparative results between iSCO and RLSA with ten times more steps on MIS. The best one is bolded.

| **MIS** | | RB-[200–300] | | RB-[800–1200] | | ER-[700–800] | | ER-[9000–11000] | |
|---|---|---|---|---|---|---|---|---|---|
| METHOD | TYPE | SIZE ↑ | TIME ↓ | SIZE ↑ | TIME ↓ | SIZE ↑ | TIME ↓ | SIZE ↑ | TIME ↓ |
| iSCO (10×) | H | 20.01 | 26.25m | 40.47 | 1.87h | 44.41 | 7.21m | 378.56 | 11.03h |
| RLSA (10×) | H | **20.10** | 6.98m | **41.83** | 10.65m | **45.05** | 2.92m | **379.19** | 17.63m |

*Table 7.* Comparative results between iSCO and RLSA with ten times more steps on MCl and MCut. The best one is bolded.

| **MCl** | | RB-[200–300] | | RB-[800–1200] | | **MCut** | | BA-[200–300] | | BA-[800–1200] | |
|---|---|---|---|---|---|---|---|---|---|---|---|
| METHOD | TYPE | SIZE ↑ | TIME ↓ | SIZE ↑ | TIME ↓ | METHOD | TYPE | SIZE ↑ | TIME ↓ | SIZE ↑ | TIME ↓ |
| iSCO (10×) | H | 18.97 | 8.81m | 40.41 | 1.83h | iSCO (10×) | H | 734.62 | 1.20h | 2960.23 | 43.98m |
| RLSA (10×) | H | 18.97 | 3.14m | **40.63** | 8.67m | RLSA (10×) | H | 734.62 | 4.07m | **2968.59** | 10.25m |

## C. Example Code for RLSA

The following Python code outlines our implementation of RLSA. The energy function corresponds to the formulas in Section 4.2 and the input parameters are summarized in Section A. In our experiments, the time measurement corresponds to the runtime of the entire RLSA function below.

```python
def energy_func(A, b, x, penalty_coeff=1.02):
    """
    The energy function is: b^Tx+penalty_coeff*x^TAx
    Return the energy and the gradient
    """

    L = A@x
    energy = torch.sum(x*(penalty_coeff*L+b), dim=0)
    grad = 2*penalty_coeff*L+b

    return energy, grad

def RLSA(graph, tau0, step_size, num_runs, num_steps, penalty_coeff):
```

```
14      """
15      graph: the graph object in torch_geometric
16      num_runs: the number of parallel SA processes
17      num_steps: the number of SA steps
18      tau0: the initial temperature
19      """
20
21      # initialization
22      num_nodes = graph.num_nodes
23
24      A = torch.sparse_coo_tensor(
25              graph.edge_index,
26              graph.edge_weight,
27              torch.Size((num_nodes, num_nodes))
28          )
29      x = torch.randint(0,2, (num_nodes, num_runs))
30
31      energy, grad = energy_func(A, graph.b, x, penalty_coeff)
32      best_energy = energy
33      best_sol = x.clone()
34
35      # SA
36      for epoch in range(num_steps):
37          # annealing
38          tau = tau0*(1-epoch/num_steps)
39
40          # sampling
41          delta = grad*(2*x-1)/2
42          term2 = -torch.kthvalue(
43                      -delta,
44                      step_size,
45                      dim=0,
46                      keepdim=True
47                  ).values
48
49          flip_prob = torch.sigmoid((delta-term2)/tau)
50          rr = torch.rand_like(x.data)
51          x = torch.where(rr<flip_prob, 1-x, x)
52
53          # update loss
54          energy, grad = energy_func(A, graph.b, x, penalty_coeff)
55          to_update = energy<best_energy
56          best_sol[:,to_update] = x[:,to_update]
57          best_energy[to_update] = energy[to_update]
58
59      return best_energy, best_sol
```

