# OpenReview forum: "Regularized Langevin Dynamics for Combinatorial Optimization"
_ICML.cc/2025/Conference — ICML 2025 poster_

### Official Review · Reviewer_B24V · 2025-03-09

**Overall Recommendation:** 3

**Summary:**

The paper introduces Regularized Langevin Dynamics, a novel sampling-based method for improving solutions to CO problems through LD. RLD incorporates regularization to control the distance of solution updates, enhancing traditional heuristics and neural network models by mitigating the issue of local optima. This approach outperforms SOTA baselines in solving large-scale graph-based CO problems, showing improved efficiency and adaptability across various settings.

**Claims And Evidence:**

- One fundamental motivation behind this paper is 'the difference in local optima between continuous optimization and combinatorial optimization.' However, it seems to lack theoretical evidence, and it remains unclear whether this difference is specific to certain function landscapes or applies to continuous/combinatorial optimization problems in general.

- Although QUBO is a fairly general formulation for CO problems, it appears to be much less effective for a prominent and extensive subset of CO problems involving global constraints, such as the TSP. The authors are encouraged to thoroughly discuss the method's limitations and scope.

**Essential References Not Discussed:**

Section 5.1 lacks an adequate discussion of recent RL-based methods [1]; the most recent work cited was published in 2022.

[1] RL4CO: an Extensive Reinforcement Learning for Combinatorial Optimization Benchmark.

**Experimental Designs Or Analyses:**

I have some concerns regarding the experimental designs:

- How do you initialize the solutions?
- The authors seem to address a fundamental problem in optimization, i.e., the trade-off between exploration and exploitation. Regularizing the update distance to a fixed scalar seems overly simplified to serve as a generic solution to this issue. I noticed that, in Tables 4 and 5, the hyperparameters are extensively tuned across problems and problem scales. Is it possible that your methods, under certain sets of hyperparameters, overfit the benchmark datasets? How do you ensure a fair comparison with the baseline methods, especially the ablation baseline (e.g., vanilla SA)? Did you also search the hyperparameters within the same ranges?
- It seems that some baselines are not directly competitive with your methods. Methods such as DIMES and DIFUSCO learn to initialize the solutions, while your sampling-based method learns to update them. Could you please elaborate on the rationale behind the choice of baselines? Is it possible to leverage the best of both worlds?
- It seems reasonable to compare your method against other approaches designed to escape local optima during solution improvements. Are there any feasible alternatives you considered for comparison?
- Is the high efficiency of your method due to the ability to use a very large batch size, given that your method is lightweight in terms of computation and memory consumption? Comparing per-instance solution duration, as discussed in many benchmarking and criticism papers, seems reasonable.

**Methods And Evaluation Criteria:**

N/A

**Other Comments Or Suggestions:**

Typos:
- "thes"; line 82
- "difference in local optimal between continuous optimization and continuous optimization", line 415

**Other Strengths And Weaknesses:**

I enjoyed reading this paper. This paper is very well-written and introduces a simple and effective method. I encourage the authors to address my concerns raised above.

**Questions For Authors:**

- Although you annotate both DIMES and RLNN as RL-type methods, can I consider them distinct types of approaches? DIMES appears to learn how to generate a solution in one step, whereas your RLNN learns to iteratively update the solution based on RLD. Furthermore, can I consider them not necessarily competitive or mutually exclusive, since you could initialize your solution with DIMES or DIFUSCO and then refine it using RLNN?
- Is it possible to adaptively regularize the solution updates?
- Is it possible to apply regularization to the solution updates only when detecting local optima?

**Relation To Broader Scientific Literature:**

This paper contributes to two threads of literature, as discussed in the Related Work section.

**Theoretical Claims:**

N/A

---

> ### Author Rebuttal · Authors · 2025-04-01
>
> > However, it seems to lack theoretical evidence
>
> >Is it possible to apply regularization to the solution updates only when detecting local optima?
>
> Thank you for your insightful question. Given the short time period, here we simply outline some perspective for the theoretical analysis.
>
>
> When not at the local optimum, RLD could accelerate the convergence. The main assumption here is that the graph is sparse (which is very common in CO), so it is unlikely for $d$ variables to be adjacent to each other and the gradient remains accurate after the change.
>
> In terms of the ability to escape optimum, we would like to point out the connection between RLD and path-auxiliary proposal [1], whose first-order version, PAFS, is also the fundamental algorithm behind iSCO. RLD could be treated as a generalized version of PAFS in terms of updating in expectation $d$ coordinates at each step, so it also satisfies Theorem 3 in that paper. Based on this generalization form, we propose a more efficient (parallel) sampling strategy to avoid the sequential sampling in PAFS, thus accelerating the search and significantly outperforming iSCO.
>
> [1] Sun, et al. Path Auxiliary Proposal for MCMC in Discrete Space. ICLR 2022.
>
> > How do you initialize the solutions?
>
> The initialization is a Bernoulli distribution of probability 0.5 for each variable.
>
> > the hyperparameters are extensively tuned across problems and problem scales
>
> We would like to clarify that all our hyperparameters are tuned via **random search** (see section A.1) rather than **grid search**. The hyperparameters in Tables 4 and 5 reflect this randomness rather than an extensive tuning process.
>
> Moreover, only the initial temperature $\tau_0$​ and the regularization target $d$ would impact performance and were carefully tuned in our experiments. We believe that tuning just two parameters is far from overfitting the dataset.
>
> > How do you ensure a fair comparison with the baseline methods, especially the ablation baseline (e.g., vanilla SA)
>
> For our ablation baseline, $K$, $T$, $\beta$ and $\tau_0$ are kept the same as the ones used in our method. Another parameter $\alpha$ is searched over {0.001,0.01,0.1}$ as presented in Figure 1.
>
> > Methods such as DIMES and DIFUSCO learn to initialize the solutions … Could you please elaborate on the rationale behind the choice of baselines?
>
> We would like to clarify that our method also initializes the solution since it does not guarantee the feasibility. We use the greedy decoding to transform the searched initialization to a feasible solution. For DIMES and DIFUSCO, they also have the greedy decoding variant. But given their poor performance, we only report the best variant with a sampling-based decoding.
>
> > Is the high efficiency of your method due to the ability to use a very large batch size … Comparing per-instance solution duration …
>
> We test all instances sequentially in our experiment, so the per-instance solution duration is simply total_time/(# instances). RLSA only needs <0.2s per instance except on ER-[9000-11000] (around 6s per instance).
>
> >  it appears to be much less effective for a prominent and extensive subset of CO problems involving global constraints
>
> This is an insightful question but we want like to mention that tackling global constraints typically need strong inductive biases and this is a common challenge for all energy-based models (EBMs). Here we refer to some recent EBM-based CO solvers that also face this dilemma and exclude the evaluation on the problems with global constraints such as TSP.
>
> [1] Haoran Sun et al., ‘Revisiting Sampling for Combinatorial Optimization’, ICML 2023.
>
> [2] Dinghuai Zhang et al., "Let the Flows Tell: Solving Graph Combinatorial Optimization Problems with GFlowNets", NeurIPS 2023 spotlight.
>
> [3] Sebastian Sanokowski et al., A Diffusion Model Framework for Unsupervised Neural Combinatorial Optimization. ICML 2024.
>
> The reason behind is because the global constraints are highly structured, while EBMs are general solvers, whose formulation itself prevents a better performance than heuristics with a strong inductive bias such as k-OPT.
>
> > It seems reasonable to compare your method against other approaches designed to escape local optima during solution improvements. Are there any feasible alternatives you considered for comparison?
>
> Note that SA itself is a strategy to escape the local optimum and we have already included the SOTA baseline iSCO. We have also experimented with the restarting strategy on top of our ablation baseline but find no clear advantage.
>
> Another method is semidefinite programming, as we included in Table 2. However, this method is in general very slow compared to SA methods.
>
>
>
>
> >Is it possible to adaptively regularize the solution updates?
>
> We have tried but found no clear gain. Note that RLD itself should be counted as an adaptive strategy on the MCMC proposal (with an adaptive $\alpha$ in Equation 8).

---

### Official Review · Reviewer_aSe5 · 2025-03-12

**Overall Recommendation:** 3

**Summary:**

*Regularized Langevin Dynamics for Combinatorial Optimization* proposes Regularized Langevin Dynamics (RLD), a sampling framework for combinatorial optimization (CO). The authors note that discrete Langevin dynamics (LD) has limitations in exploration when applied to CO. RLD addresses this by enforcing an expected distance between sampled and current solutions, avoiding local minima.

Two CO solvers are developed based on RLD: Regularized Langevin Simulated Annealing (RLSA) and Regularized Langevin Neural Network (RLNN). RLSA is a simple yet effective algorithm that often outperforms previous state - of - the - art (SOTA) SA methods. For example, it reduces the running time of the previous SOTA SA method by up to 80% while achieving equal or better performance. RLNN, partially based on reinforcement learning, can be trained efficiently with a local objective.

Empirical results on three classical CO problems (Maximum Independent Set, Maximum Clique, and Max Cut) show that both RLSA and RLNN can achieve comparable or better performance than previous SOTA SA and NN - based solvers. Ablation studies verify the effectiveness of the regularization in RLD. Overall, RLD offers a promising framework for enhancing traditional heuristics and NN models to solve CO problems, though its effectiveness on non - binary data and theoretical understanding remain areas for future research.

**Claims And Evidence:**

The paper claims that regularization could mitigate the local optima during the optimization process. It then explains this claim using an MIS problem as an example as well as experiments. It could be better if  provide more theoretical analysis.

**Essential References Not Discussed:**

The paper does not contain missed references as far as I know.

**Experimental Designs Or Analyses:**

I checked the experimental results, the ablation study and the curve of primal gap, which gives convincing results to back up the proposed method.

**Methods And Evaluation Criteria:**

The method and evaluation make sense in general.

**Other Comments Or Suggestions:**

None

**Other Strengths And Weaknesses:**

Advantage:
- The proposed RLSA and RLNN achives better decision quality with lower inference time. The ablation study validates the effectiveness of proposed regularization.
- The proposed regularization contributions to the discrete sampling algorithm for addressing the local optimal problem during optimization.

Disadvantage:
- It would be better if the paper could give a theoretical analysis of the proposed regularization.

**Questions For Authors:**

Questions:
- Is there any advantage of RLNN compared to RLSA? It seems it falls behind of RLSA on both decision quatlity and inference time.

**Relation To Broader Scientific Literature:**

The paper is related to combinatorial optimization and gradient-guided generative algorithms.

**Theoretical Claims:**

The paper does not give theoretical claims.

---

> ### Author Rebuttal · Authors · 2025-04-01
>
> We sincerely thank you for your positive feedback, here are the response to your concerns.
>
> >Is there any advantage of RLNN compared to RLSA? It seems it falls behind of RLSA on both decision quality and inference time.
>
> Recall that our RLNN is designed to address a limitation of RLSA, i.e., its reliance on closed-form formulation of the gradient (as mentioned in the first sentence of Section 3.3).  In other words, RLSA requires additional knowledge for making it works; when such knowledge is not available, RLNN is a better alternative. And it should not be surprising that RLSA performs better since it has more prior knowledge.
>
>
> > It would be better if the paper could give a theoretical analysis of the proposed regularization.
>
> Thank you for your suggestion. Given the short time period, here we simply outline some perspective for the theoretical analysis.
>
>
> When not at the local optimum, RLD could accelerate the convergence. The main assumption here is that the graph is sparse, so it is unlikely for $d$ variables to be adjacent to each other and the gradient remains accurate after the change.
>
> In terms of the ability to escape optimum, we would like to point out the connection between RLD and path-auxiliary proposal [1], whose first-order version, PAFS, is also the fundamental algorithm behind iSCO. RLD could be treated as a generalized version of PAFS in terms of updating in expectation $d$ coordinates at each step, so it also satisfies Theorem 3 in that paper. Based on this generalization form, we propose a more efficient (parallel) sampling strategy to avoid the sequential sampling in PAFS, thus accelerating the search and significantly outperforming iSCO.
>
> [1] Sun, et al. Path Auxiliary Proposal for MCMC in Discrete Space. ICLR 2022.

---

### Official Review · Reviewer_pJpR · 2025-03-12

**Overall Recommendation:** 3

**Summary:**

This paper introduces Regularized Langevin Dynamics (RLD), an approach inspired by normalized gradient descent to enhance combinatorial optimization (CO) methods. The authors develop two specific algorithms: Regularized Langevin Simulated Annealing (RLSA), which incorporates simulated annealing, and Regularized Langevin Neural Network (RLNN), which integrates neural networks. Through empirical evaluations on Maximum Independent Set (MIS), Maximum Clique (MCl), and Max Cut (MCut) problems, the authors demonstrate that their regularization technique improves performance over unregularized counterparts. While RLNN provides moderate improvements over certain learning-based methods, RLSA achieves performance comparable to the current state-of-the-art heuristic, iSCO, in CO problems. The authors further conduct ablation studies to validate the efficacy of the regularization.

## update after rebuttal
I increase my score to 3, but the writing should be further improved.

**Claims And Evidence:**

While the paper presents strong empirical evidence to support its claims, concerns remain regarding the fairness of experimental comparisons. Specifically, the evaluation methodology for iSCO may not be entirely consistent, as discussed in the weaknesses section.

**Essential References Not Discussed:**

There are no critical omissions in terms of references.

**Experimental Designs Or Analyses:**

There are concerns regarding experimental fairness, particularly in the comparison of iSCO and RLSA, as discussed in the weaknesses section. A more rigorous comparison based on computational time rather than fixed iterations would be preferable. Additionally, the omission of Parallel Quasi-Quantum Annealing (PQQA) as a baseline is a notable limitation.

**Reference**

Ichikawa Y, Arai Y. Optimization by Parallel Quasi-Quantum Annealing with Gradient-Based Sampling. arXiv preprint arXiv:2409.02135, 2024.

**Methods And Evaluation Criteria:**

The proposed methods and evaluation criteria are generally appropriate for CO problems. However, the choice of evaluation metrics for MIS could be improved. Metrics such as average performance drop (Drop) or approximation ratio (ApR), as used in prior works (e.g., Qiu et al., 2022; Ichikawa & Arai, 2024), might provide a more comprehensive assessment of performance.

**Reference**

Qiu R, Sun Z, Yang Y. DIMES: A Differentiable Meta Solver for Combinatorial Optimization Problems. NeurIPS, 2022.

Ichikawa Y, Arai Y. Optimization by Parallel Quasi-Quantum Annealing with Gradient-Based Sampling. arXiv preprint arXiv:2409.02135, 2024.

**Other Comments Or Suggestions:**

Below is a refined list of typos and minor textual issues:

1. Line 82 (Left Column): Change "thes" to "the."

2. Line 150 (Left Column): Change "an" to "a."

3. Line 178 (Right Column): Change "coefficiednt" to "coefficient."

4. Line 219 (Left Column): Change "mean-filed" to "mean-field."

5. Line 227 (Right Column): Change "Barabasi-Albert" to "Barabási-Albert."

6. Line 263 (Right Column): Change "mean-filed" to "mean-field."

7. Line 724: Capitalize "On."

8. Equation (13): Replace "$-c^{T}x_{i}$" with "$-c^{T}x$" for consistency.

9. Line 703: Correct the summation notation: "$D_{ii} = \sum j=1^N \hat{A}{i,j}$" should be "$D{ii} = \sum_{j=1}^N \hat{A}_{i,j}$".

**Other Strengths And Weaknesses:**

**Strengths:**

1. The concept of regularization in Langevin dynamics is well-motivated and empirically validated.

2. The implementation is simple yet effective, making it accessible for further research and applications.

**Weaknesses:**

1. **Experimental Fairness**: The experimental section in the iSCO paper reports DIMES achieving a size of 42.06 on ER-[700–800] in 12.01 minutes and 332.80 on ER-[9000–11000] in 12.51 minutes, which aligns with previously reported results. However, the iSCO results in this paper do not consistently reflect this. Additionally, the claim that “For a fair comparison, we run iSCO with the same number of steps and trials as we did for RLSA” is questionable. A more appropriate fairness criterion would be to impose the same computational time limit. For instance, iSCO (using fewer steps) reaches a size of 44.77 in 1.38 minutes, whereas RLNN takes 1.37 minutes to reach a size of 43.34. A time-based comparison would be fairer.

2. **Experimental Performance**: In heuristic methods, iSCO outperforms RLSA in both the fewer-step scenario (size = 44.77) and the more-steps scenario (size = 45.15) compared to RLSA’s result (size = 44.10). In neural network approaches, DiffUCO achieves a size of 38.87 in 4.95 minutes on RB-[800–1200], whereas RLNN takes 6.24 minutes to reach a size of 38.46, indicating that RLNN is weaker than DiffUCO. Also, there are too many missing values (“-”) in the experimental tables, making cross-comparison difficult.

3. **Significance of the Neural Network Approach**: Since RLSA consistently outperforms RLNN across test cases, the contribution of the neural network method appears to be limited.

4. **Presentation Issues**: The presentation needs to be improved, and the paper contains an excessive number of typographical errors.


**Reference**

Sun H, Goshvadi K, Nova A, et al. Revisiting Sampling for Combinatorial Optimization. ICML, 2023.

**Questions For Authors:**

1. **Evaluation Criteria Consistency**: Shouldn’t the evaluation criteria be aligned across all experiments to ensure fairness, given that Sun et al. (2023) reports consistent DIMES results when comparing with iSCO?

2. **Iteration Step Size Justification**: Wouldn’t comparing performance at multiple time budgets rather than fixing iteration steps arbitrarily be a more rigorous approach, given that iSCO can reach better solutions within shorter execution times?

**Relation To Broader Scientific Literature:**

The paper is related to gradient-based sampling and combinatorial optimization.

**Theoretical Claims:**

The theoretical claims appear correct, and I did not find any issues in the proofs presented. The mathematical formulations align with previous works on Langevin dynamics and simulated annealing.

---

> ### Author Rebuttal · Authors · 2025-04-01
>
> > There are concerns regarding experimental fairness, particularly in the comparison of iSCO and RLSA… The experimental section in the iSCO paper reports DIMES achieving a size of 42.06 on ER-[700–800] in 12.01 minutes and 332.80 on ER-[9000–11000] in 12.51 minutes, which aligns with previously reported results. However, the iSCO results in this paper do not consistently reflect this…
>
> We  sincerely appreciate  your question and want to offer an important clarification on a confounding issue. That is, in the  iSCO paper (Sun et al. 2023), they reported the **average running time** per instance for iSCO, while they reported the **total running time** for all other methods (DIMES included). This mis-alignment in time measurements makes iSCO look 16 or 128 times better than the reality (given the test-set size is 16 and 100 for ER graphs).   We have contacted the authors of that paper, and they confirmed that it was indeed an error on their behalf. (if needed and possible, we could attach their relevant material)
>
> To fix such a confounding issue, we reported the **total running time** for all the methods in our results tables.  This is why our numbers look inconsistent to that in the iSCO paper, but for a good reason. We will revise related sentences about those tables, to make the point more transparent.
>
> We fully understand that the mistake in Sun et al. 2023 caused the confusion in your review.  If the above clarification sufficiently addressed this issue, would you kindly adjust your score accordingly?
>
> > Since RLSA consistently outperforms RLNN across test cases, the contribution of the neural network method appears to be limited.
>
> Recall that our RLNN is designed to address a limitation of RLSA, i.e., its reliance on closed-form formulation of the gradient (as mentioned in the first sentence of Section 3.3).  In other words, RLSA requires additional knowledge for making it works; when such knowledge is not available, RLNN is a better alternative. And it should not be surprising that RLSA performs better since it has more prior knowledge.
>
>
>
> > RLNN is weaker than DiffUCO
>
> In fact, RLNN has comparable performance to DiffUCO, based on the results in Tables 1 and 2 (RLNN outperforms DiffUCO in 4 out of 6 datasets). More importantly, the training of RLNN is much more efficient than that of DiffUCO because RLNN only needs to be trained locally while DiffUCO relies on a sequential training loss.  For example, the training time is 1 hour for RLNN and 2 days plus 2 hours  for DiffUCO (reported in section C.6, Sanokowski, et al. 2024)  on RB-200.
>
> > typos & metrics
>
> We want to apologize for the presentation issues, and we would fix them in our future version.

---

### Official Review · Reviewer_ZFeK · 2025-03-13

**Overall Recommendation:** 2

**Summary:**

This paper proposes improvements to existing diffusion-based amortized neural samplers and discrete Langevin dynamics samplers for combinatorial optimization, introducing simple regularization techniques aimed at mitigating issues with local minima in discrete spaces. The key claim is that avoiding local minima in discrete optimization is inherently more challenging than in continuous optimization, necessitating additional regularization methods. While the idea is intuitive and the proposed methods (Regularized Langevin Simulated Annealing and Regularized Langevin Neural Network) are straightforward and well-presented, the novelty appears somewhat incremental—essentially amounting to relatively simple adjustments to existing techniques. Empirical evaluations on standard combinatorial optimization benchmarks do demonstrate improved performance and computational efficiency; however, the overall methodological innovation is limited, as it mainly builds upon existing discrete Langevin dynamics frameworks with slight modifications.

**Claims And Evidence:**

The main claim of the paper—that avoiding local minima in discrete optimization is inherently more challenging compared to continuous optimization and thus necessitates more sophisticated regularization techniques—is reasonable and well-motivated. The authors provide clear empirical evidence demonstrating that their proposed regularization approach significantly improves performance compared to standard discrete Langevin dynamics. Specifically, their experiments convincingly show faster convergence and better-quality solutions on multiple combinatorial optimization benchmarks. However, the theoretical justification for why their specific form of regularization effectively escapes local minima remains somewhat intuitive rather than rigorously proven. Although the experimental results support their main claims, further theoretical analysis would strengthen their argument regarding the superiority of the proposed regularization method over existing approaches.

**Essential References Not Discussed:**

They have to includes some diffusion sampler works, amortized inference works and RL for vehicle routing works.

Missed critical paper:

[1] Dinghuai Zhang et al., "Let the Flows Tell: Solving Graph Combinatorial Optimization Problems with GFlowNets", NeurIPS 2023 spotlight.

**Experimental Designs Or Analyses:**

Already noted in Methods And Evaluation Criteria.

**Methods And Evaluation Criteria:**

The authors' choice of benchmarks is reasonable and aligns well with common evaluation practices in combinatorial optimization. They use standard problems (e.g., maximum independent set, max clique, and max cut), which primarily involve local constraints. However, previous diffusion-based sampling methods have also demonstrated applicability to problems with global constraints, such as vehicle routing or scheduling tasks. Given that these tasks require satisfying complex global constraints—significantly more challenging than local constraints like those present in the current benchmarks—it remains an open question whether the proposed regularization method would generalize well to such problems. Specifically, it would be insightful to see whether the method can effectively handle combinatorial optimization tasks involving intricate global constraint matching, which is known to pose substantial difficulties for sampling-based methods. Exploring applications such as vehicle routing or scheduling would further clarify the strengths and limitations of the proposed framework.

**Other Comments Or Suggestions:**

No.

**Other Strengths And Weaknesses:**

No.

**Questions For Authors:**

1. The proposed sampler demonstrates good performance on medium-scale combinatorial optimization benchmarks. Have you evaluated its performance on larger-scale problems (e.g., tens of thousands of nodes)? If so, how does scalability and performance stability compare to baseline methods at those larger scales?

2. Considering that your evaluations mainly focus on combinatorial problems with relatively simple (local) constraints, have you investigated how well your method performs on problems with stronger global constraints, such as the Traveling Salesman Problem (TSP)? I'm particularly curious whether your regularization approach can effectively handle the global constraints inherent to routing or scheduling problems.

**Relation To Broader Scientific Literature:**

This discrete sampling method can potentially applied to impactful combinatorial tasks such as LLM reasoning (EBM style) and molecular dynamics simulation.

**Theoretical Claims:**

No theory here.

---

> ### Author Rebuttal · Authors · 2025-04-01
>
> We sincerely appreciate the reviewer's thoughtful comments and constructive suggestions.
>
> > the novelty appears somewhat incremental—essentially amounting to relatively simple adjustments to existing techniques
>
>
>
> We see your point but would like to highlight that our approach addresses a fundamental limitation of the previous method (discrete Langevin-like sampling in Zhang et al., 2022, and iSCO in Sun et al. 2023) in solving CO problems, with SOTA performance on benchmarks and significantly reduced computational costs.  Which should be informative for the research community, we hope.
>
>
> > However, the theoretical justification for why their specific form of regularization effectively escapes local minima remains somewhat intuitive rather than rigorously proven.
>
> Thank you for your insightful question. Given the short time period, here we simply outline some perspective for the theoretical analysis.
>
>
> When not at the local optimum, RLD could accelerate the convergence. The main assumption here is that the graph is sparse, so it is unlikely for d variables to be adjacent to each other and the gradient remains accurate after the change.
>
> In terms of the ability to escape optimum, we would like to point out the connection between RLD and path-auxiliary proposal [1], whose first-order version, PAFS, is also the fundamental algorithm behind iSCO. RLD could be treated as a generalized version of PAFS in terms of updating in expectation $d$ coordinates at each step, so it also satisfies Theorem 3 in that paper. Based on this generalization form, we propose a more efficient (parallel) sampling strategy to avoid the sequential sampling in PAFS, thus accelerating the search and significantly outperforming iSCO.
>
>
>
> [1] Sun, et al. Path Auxiliary Proposal for MCMC in Discrete Space. ICLR 2022.
>
> > Have you evaluated its performance on larger-scale problems (e.g., tens of thousands of nodes)?
>
> Please notice our results on ER-[9000-11000] in Table 1: RLSA outperforms iSCO using only 2.5% running time. Also, RLNN takes a similar amount of time as DIMES but delivers significantly better results. All these results provide strong evidence for the advantage of our approach on larger-scale problems.
>
>
> > However, previous diffusion-based sampling methods have also demonstrated applicability to problems with global constraints, such as vehicle routing or scheduling tasks.
> > Considering that your evaluations mainly focus on combinatorial problems with relatively simple (local) constraints, have you investigated how well your method performs on problems with stronger global constraints, such as the Traveling Salesman Problem (TSP)?
>
> This is an insightful question but we would like to clarify that tackling global constraints typically need strong inductive biases and this is a common challenge for all energy-based models (EBMs). Here we refer to some recent EBM-based CO solvers that also face this dilemma and exclude the evaluation on TSP.
>
> [1] Haoran Sun et al., ‘Revisiting Sampling for Combinatorial Optimization’, ICML 2023.
>
> [2] Dinghuai Zhang et al., "Let the Flows Tell: Solving Graph Combinatorial Optimization Problems with GFlowNets", NeurIPS 2023 spotlight.
>
> [3] Sebastian Sanokowski et al., A Diffusion Model Framework for Unsupervised Neural Combinatorial Optimization. ICML 2024.
>
> The reason behind is because global constraints are highly structured, while EBMs are general solvers, whose formulation itself prevents a better performance than heuristics with a strong inductive bias such as k-OPT.
>
> Besides, we would like to mention that diffusion models themselves cannot handle global constraints, whose success heavily relies on (i) the supervision, which smooths the landscape (ii) decoding heuristics (2-OPT, MCTS or permutation), which is the key to handle the global constraint.

---

### Decision · Program_Chairs · 2025-05-01

**Decision:**

Accept (poster)

**Comment:**

The reviewers praise the idea of regularizing Langevin dynamics and applaud the simplicity of the approach as well as the effectiveness. Some reviewers argue that the novelty is too low, but I think simplicity is sometimes confused with low novelty and thus propose ignoring these comments entirely, as simplicity is good. I have also assessed the experiments due to some questions from the reviewers, but believe them to be adequate for showing the performance of the method. The baselines presented are also up to date. The final negative remark is regarding the clarity of the paper. I encourage the authors to clean up their descriptions according to the reviewer's comments and I believe the paper will be in an adequate state for publication.